# Effects of Intravenous Antimicrobial Drugs on the Equine Fecal Microbiome

**DOI:** 10.3390/ani12081013

**Published:** 2022-04-13

**Authors:** Rachel S. Liepman, Jacob M. Swink, Greg G. Habing, Prosper N. Boyaka, Benjamin Caddey, Marcio Costa, Diego E. Gomez, Ramiro E. Toribio

**Affiliations:** 1Department of Veterinary Clinical Sciences, College of Veterinary Medicine, The Ohio State University, Columbus, OH 43210, USA; drliep@gmail.com (R.S.L.); swink.4@osu.edu (J.M.S.); 2Department of Preventive Medicine, College of Veterinary Medicine, The Ohio State University, Columbus, OH 43210, USA; habing.4@osu.edu; 3Department of Veterinary Biosciences, College of Veterinary Medicine, The Ohio State University, Columbus, OH 43210, USA; boyaka.1@osu.edu; 4Department of Production Animal Health, Faculty of Veterinary Medicine, University of Calgary, Calgary, AB T2N 1N4, Canada; benjamin.caddey@ucalgary.ca; 5Department of Veterinary Biomedical Sciences, Faculté de Médecine Vétérinaire, University of Montreal, Saint Hyacinthe, QC J2S 2M2, Canada; marcio.costa@umontreal.ca; 6Department of Clinical Studies, Ontario Veterinary College, University of Guelph, Guelph, ON N1G 2W1, Canada; dgomezni@uoguelph.ca

**Keywords:** horse, intestine, gastrointestinal, antibiotics, stewardship, microbiota, Lachnospiraceae, Verrucomicrobia, Clostridia

## Abstract

**Simple Summary:**

The objectives of this study were to determine the effect of intravenous administration of antimicrobials (ceftiofur, enrofloxacin, oxytetracycline) on equine fecal bacterial communities over time. Healthy horses were treated for 5 days with enrofloxacin, ceftiofur sodium, oxytetracycline and saline solution, and fecal samples were collected over 30 days. Microbiome analysis was carried out via 16S rRNA gene sequencing. Microbial diversity and abundance were altered using ceftiofur and enrofloxacin. This study showed that antimicrobials alter gut bacterial communities, which could predispose horses to gastrointestinal inflammation, diarrhea and possibly systemic disorders.

**Abstract:**

Alterations in the gastrointestinal microbiota after antimicrobial therapy in horses can result in loss of colonization resistance and changes in bacterial metabolic function. It is hypothesized that these changes facilitate gastrointestinal inflammation, pathogen expansion and the development of diarrhea. The objectives of this study were to determine the effect of intravenous administration of antimicrobial drugs (ceftiofur, enrofloxacin, oxytetracycline) on equine fecal bacterial communities over time, to investigate whether those changes are detectable after 5 days of treatment and whether they persist over time (30 days). Sixteen horses were randomly assigned into 4 treatment groups: group 1 (enrofloxacin, n = 4); group 2 (ceftiofur sodium, n = 4); group 3 (oxytetracycline, n = 4); group 4 (0.9% saline solution, placebo, n = 4). Antimicrobial therapy was administered for 5 days. Fecal samples were obtained before (day 0) and at 3, 5 and 30 days of the study period. Bacterial DNA was amplified using specific primers to the hypervariable region V1–V3 of the 16S rRNA gene using a 454 FLX-Titanium pyrosequencer. Antimicrobial therapy failed to cause any changes in physical examination parameters, behavior, appetite or fecal output or consistency throughout the study in any horse. There was a significant effect of treatment on alpha diversity indices (richness) over the treatment interval for ceftiofur on days 0 vs. 3 (*p* < 0.05), but not for other antimicrobials (*p* > 0.05). Microbial composition was significantly different (*p* < 0.05) across treatment group and day, but not for interactions between treatment and day, regardless of taxonomic level and beta-diversity distance metric. The most significant antimicrobial effects on relative abundance were noted after intravenous administration of ceftiofur and enrofloxacin. The relative abundance of Fibrobacteres was markedly lower on day 3 compared to other days in the ceftiofur and enrofloxacin treatment groups. There was an increase in Clostridia and Lachnospiraceae from day 0 to days 3 and 5 in ceftiofur and enrofloxacin treated groups. These findings showed the negative effect of antimicrobial drugs on bacterial communities associated with gut health (Fibrobacteres and Lachnospiraceae) and indicate that changes in specific taxa could predispose horses to gastrointestinal inflammation and the development of diarrhea.

## 1. Introduction

Antimicrobial therapy is implicated in antimicrobial-associated diarrhea (AAD) in horses and humans [1,2,3,4,5,6]. In horses, ADD often leads to serious complications including sepsis, laminitis, multiorgan failure, and death [7]. It is hypothesized that disruption of the enteric microbiota after antimicrobial therapy in horses leads to loss of colonization resistance and alteration of the microbial metabolic function which favors gastrointestinal inflammation and pathogen proliferation [6,8]. In the research setting, using rodent models of AAD, certain antimicrobial drugs are used to overcome the inherent GI microbiota and allow for pathogenic microorganisms to proliferate and cause disease [9,10]. Administration of ciprofloxacin in humans and tulathromycin in calves causes significant changes in the fecal microbiota, both short- and long-term [11,12]. In horses, administration of antimicrobial drugs modifies the fecal microbiota of healthy horses shortly after initiation of therapy, but fecal microbiota returns to pre-treatment profile approximately 25 days after cessation of antimicrobial treatment [13]. However, specific differences in community membership persist, especially in response to potentiated sulfonamide administration [14,15]. These findings support the theory that the GI microbiota in many mammalian species is affected both short- and long-term by treatment with different antimicrobial drugs. Although evidence exists that antimicrobial drugs alter the bacterial population of the equine GI tract [13,16,17], there is limited information on how it is altered, the length of time needed to observe such changes, how long alterations persist and which organisms are the most affected.

Antimicrobial drugs associated with AAD in horses and commonly used in equine practice include tetracyclines, ceftiofur, enrofloxacin and potentiated sulfonamides [4,13]. These antimicrobial drugs have different mechanisms of action and spectra of activity. Oxytetracycline is a tetracycline antimicrobial drug that inhibits bacterial protein synthesis, is bacteriostatic and has a broad-spectrum of activity. Ceftiofur is a third-generation cephalosporin that inhibits bacterial cell wall synthesis, is bactericidal and has a broad-spectrum of activity. Enrofloxacin is a fluoroquinolone antimicrobial drug that interferes with bacterial DNA metabolism, is bactericidal and has a broad-spectrum of activity, especially against bacteria from the Enterobacteriaceae family.

The objectives of this study were to determine how intravenous oxytetracycline, ceftiofur and enrofloxacin modify equine fecal bacterial communities over time, to investigate whether those changes are detectable after 5 days of treatment, and whether they persist over time (30 days). We hypothesized that diversity of the equine fecal microbiome will be reduced by these antimicrobial drugs compared to placebo treatment and that 5 days of treatment will cause significant alterations, some of which will persist over a 30-day period post-treatment.

## 2. Materials and Methods

### 2.1. Animals

Sixteen healthy horses of the teaching and research herd of The Ohio State University were used for this study with a mean and median age of 13.9 and 14 years old, respectively (standard deviation 4.2 years, range 8–24 years old). Five mares and 11 geldings were used, and breeds represented included 5 Thoroughbreds, 5 Warmbloods, 3 Quarter horses, 2 Standardbreds, 1 Saddlebred and 1 Appaloosa.

All horses were housed at the same location under standardized conditions and fed the same diet of grass hay for 3 weeks prior to study inclusion. Horses were considered healthy based upon physical examination, hematology, serum chemistry and fibrinogen concentrations. Horses had no evidence of endoparasitism based on examination of fecal samples. All horses were free of known GI disease, had no history of antimicrobial administration for at least 6 months prior to the study and were up to date on core vaccinations and deworming. The Ohio State University Institutional Animal Care and Use Committee approved this study.

Horses were randomly assigned into 4 treatment groups: group 1 (enrofloxacin, n = 4); group 2 (ceftiofur sodium, n = 4); group 3 (oxytetracycline, n = 4); group 4 (0.9% saline solution, placebo, n = 4). All treatments and sampling were performed during the same time of year under the same conditions in all horses by two investigators (RL, JS).

### 2.2. Experiment

A 14 gauge 5.25 inch (13 cm) polyurethane catheter (Mila International, Erlanger, KY, USA) was aseptically inserted in the jugular vein of each horse for antibiotic administration. Blood was drawn from each horse for complete blood count and serum chemistry analysis before treatment and at the end of the treatment period. Horses were administered enrofloxacin (Baytril, Bayer Animal Health, Shawnee, KS, USA; 7.5 mg/kg, IV, q24h [AM] and 30 mL of 0.9% sodium chloride solution, IV, q24h [PM]), ceftiofur sodium (Naxcel; Zoetis, Florham Park, NJ, USA; 2.2 mg/kg, IV, q12h), oxytetracycline (Oxytetracycline Injection 200, Norbrook Inc., Lanexa, KS, USA; 6.6 mg/kg, IV, q24h and 30 mL of 0.9% sodium chloride solution, IV, q24h [PM]) and 0.9% sodium chloride solution (Baxter, Deerfield, IL, USA; 30 mL, IV, q12h) for 5 days. Physical examinations (heart rate, respiratory rate, mucous membrane appearance, capillary refill time, digital pulses, abdominal auscultation and rectal temperature) and evaluation of fecal output and consistency were performed twice daily. Antimicrobial drugs were administered after fecal samples were obtained. Fecal samples were collected from the rectum of each horse via a sterile rectal sleeve every morning, frozen in liquid nitrogen and subsequently stored at −80 °C until processing.

### 2.3. DNA Extraction, PCR Amplification and Sequencing

Samples from baseline (time 0, prior to treatment), 1, 3, 5 and 30 days post-treatment were analyzed. Bacterial DNA was isolated from fecal samples using a commercial kit (QIAamp DNA Stool Mini Kit, QIAGEN, Valencia, CA, USA). DNA quantity and quality (260/280 ratio) was determined via spectrophotometry (NanoDrop, Thermo Scientific, Wilmington, DE, USA).

Bacterial DNA was amplified using specific primers to the hypervariable region V1-V3 of the 16S rRNA gene using a 454 FLX-Titanium pyrosequencer (Roche, Branford, CT, USA) [18]. An approximately 500 bp fragment of the 16S rRNA gene was amplified (HotStart Master Mix Kit, QIAGEN) using 100 ng of DNA and eubacterial primers specific for most GI bacteria and numbered in relation to the *Escherichia coli* 16S rRNA gene (28F = 5′-GAGTTTGATCNTGGCTCAG-3′; 518R = 5′-GTNTTACNGCGGCKGCTG-3′). The forward primer carried the A pyrosequencing adaptor and a multiplex identifier (MID) sequence, while the reverse primer carried the B pyrosequencing adaptor. The following cycling conditions were used: denaturation at 94 °C for 3 min, followed by 32 cycles of 94 °C for 30 s, annealing at 60 °C for 40 s and 72 °C for 1 min; and a final elongation step at 72 °C for 5 min. A secondary PCR was performed to incorporate linker tags as described for multiplexed 454 FLX amplicon pyrosequencing (Roche, Branford, CT, USA). Amplified PCR products were purified using Ampure beads (Beckman Coulter, Indianapolis, IN, USA).

## 3. Data Analysis

The software Mothur [19] was used to cluster sequences into operational taxonomic units (OTU) assignments (97% similarity) after adaptor and MID removal, nucleotide trimming and discarding fragments of <200 base pairs. Sequences with ambiguous calls were excluded from the analysis. The UChime program was used to detect and exclude potential chimeras. Alignments were made using the SILVA database (http://www.arb-silva.de, accessed on 15 July 2021) as reference and the RDP database was used for taxonomic classification. All OTUs belonging to the same genus were clustered for further analysis (phylotypes). Random subsampling was performed to avoid bias caused by uneven samples using the smallest number of reads found among all samples.

Operational taxonomic unit counts for each sample were normalized via total sum scaling (relative abundance transformation). Relative abundances were calculated on all samples as the percent of taxonomically classified OTUs at each taxonomic level. Alpha diversity was calculated using Simpson’s (diversity), Chao-1 (richness) and Shannon’s evenness indices (evenness). A generalized linear mixed model was used to investigate differences within and between groups. The dependent variables for time, treatment and their interaction were forced into each model and the model residuals subjectively assessed for normality. To account for repeated sampling of the same horses over time, horse was included as a repeated statement.

Principal coordinate analysis (PCoA) and permutational multivariate analysis of variance (PERMANOVA) were performed on beta-diversity metrics (Bray-Curtis and Jaccard distances) at phylum, family and genus level taxonomic grouping to visualize differences in microbial composition among treatment group and day. PERMANOVAs had 9999 permutations stratified by horse to determine significant differences in microbial composition across treatment group, day and treatment-day interactions. Pairwise differences on beta-diversity distances for treatment-day interactions were analyzed using Bray-Curtis distances at genus level taxonomy. Linear mixed models were built with pairwise distance as the outcome, treatment and day with interactions as fixed effects, horse as a random effect and all pairwise p-values being adjusted using Tukey’s HSD. Beta-diversity metrics (Bray-Curtis and Jaccard distances) and PERMANOVAs were calculated using the vegan R package (v. 2.5.7), PCoAs were performed using the ape R package (v. 5.5), and linear mixed models were analyzed using lme4 (v. 1.1.23) and emmeans (1.6.0) R packages. Data analysis was performed using R software v. 3.6.3 (https://www.r-project.org, accessed on 1 March 2022), and *p*-values less than 0.05 were considered statistically significant.

## 4. Results

### 4.1. Clinical Response to Antimicrobial Therapy

There were no changes in physical examination parameters, behavior, appetite, fecal output or fecal consistency throughout the study in any horse, and none of the horses developed clinical evidence of diarrhea during the study period. Baseline complete blood count and serum chemistry profiles were within normal limits during this time.

### 4.2. Overall Assessment of the Sequences

An average of 5132 ± 1639 reads per sample were used for taxonomic classification and subsampling was performed at 2000 reads per sample for alpha and beta diversity analysis. Sequences were assigned to 18 phyla, 34 classes, 55 orders, 87 families and 151 genera. An average of 19.9% of sequences remained unclassified at the phylum level. Good’s coverage estimates (mean 97%, standard deviation 0.5%) and rarefaction curves after subsampling verified adequate coverage of diversity in all samples (data not shown).

### 4.3. Alpha Diversity

Overall, there was no significant effect of treatment or time on alpha diversity indices over the treatment interval in the studied population of healthy horses. Genus-level OTU richness for the ceftiofur treatment group was significantly greater (*p* < 0.05) on day 0 compared to day 3 (Figure 1). No additional statistically significant differences in richness estimates were observed for the other treatment groups, although ceftiofur, enrofloxacin, and oxytetracycline treatment groups followed similar richness trends with richness being highest at day 0 and lowest on day 3 before increasing by day 30 (Figure 1).

### 4.4. Beta Diversity

Microbial composition was significantly different (*p* < 0.05) across treatment group and day, but not for interactions between treatment and day, regardless of taxonomic level and beta-diversity distance metric (Figure 2A,B). Bray-Curtis distances consistently captured more variation in microbial composition compared to Jaccard distances, regardless of taxonomic level analyzed (Figure 2A,B). At the family and genus-level, PERMANOVA R^2^ estimates for Bray-Curtis distances indicated that day and day-treatment interactions explain 23% and 15% of variation in microbial composition, respectively (Figure 2A). Principal coordinate analysis showed signs of clustering at genus-level grouping for Bray-Curtis distances in which enrofloxacin, oxytetracycline and saline samples at days 0, 3 and 5 cluster together (Figure 2A). Samples from day 30 clustered together, but clustering was not observed for the remaining treatment groups (Figure 2A). Pairwise Bray-Curtis differences only identified significant differences in microbial composition between treatments on day 3 and day 5 (Figure 3). On day 3, the microbial composition of saline treated animals was significantly more similar (*p* < 0.05) to oxytetracycline and enrofloxacin treated groups compared to ceftiofur (Figure 3). On day 5, saline treated animals were only significantly more similar (*p* < 0.05) to oxytetracycline than to ceftiofur treated horses (Figure 3).

### 4.5. Relative Abundance

Four main phyla were highly prevalent throughout all treatment groups, Firmicutes, Bacteroidetes, Fibrobacteres and Proteobacteria, with all other phyla being present at <1% relative abundance (Figure 4). In ceftiofur and enrofloxacin treatment groups, relative abundance of Fibrobacteres was lower on day 3 compared to other days measured (Figure 4). At genus level taxonomy, larger differences in microbial population dynamics become apparent during treatment progression. Relative abundance of an unclassified genus of the phylum Bacteroidetes remains relatively constant throughout treatment progression, except for ceftiofur, in which relative abundance decreased on days 3 and 5 (Figure 4). Relative abundance of an unclassified genus of the family Lachnospiraceae increased on day 3 for all treatments except saline, where this taxon remained consistently low before increasing on day 30 (Figure 4). *Herbaspirillum* was highly abundant on day 0, before lowering to <1% relative abundance for all treatment groups during the study period (Figure 5). Clostridia populations increased in relative abundance from day 0 to days 3 and 5 for ceftiofur and enrofloxacin treated groups, before returning to baseline levels by day 30 (Figure 5). Clostridia relative abundance did not appear to vary by day for oxytetracycline and saline groups (Figure 5). *Fibrobacter* slowly decreased over the 30-day period for saline and oxytetracycline treated groups but lowered to <1% relative abundance in ceftiofur and enrofloxacin treated groups by day 3 (Figure 5).

## 5. Discussion

The fecal microbiota of healthy horses in our population was similar to that reported in other studies, with Firmicutes and Bacteroidetes dominating and Proteobacteria and *Fibrobacter* being less abundant [20,21,22,23,24,25]. Verrucomicrobia has been reported as a major phylum in several equine fecal microbiota studies [26,27,28]; however, this phylum was a minor component in our population of horses, likely because of methodological differences related to PCR amplification [15,22]. These differences can also be explained by dissimilarities in the signalment, management practices and environmental conditions in which our horses were maintained compared to previous studies.

The most significant antimicrobial effects were noted after intravenous administration of ceftiofur and enrofloxacin. Only minor changes were seen with oxytetracycline compared to the saline group. All three antimicrobials have broad spectra of activity and have some degree of gastrointestinal excretion; therefore, an impact on gut microbiota was expected. In our study, the relative abundance of Fibrobacteres was markedly lower on day 3 compared to other days in the ceftiofur and enrofloxacin treatment groups. This is of interest because horses with AAD had a reduction in the relative abundance of the Fibrobacteraceae family [16,29] in horses receiving metronidazole orally [17]. Additionally, a lower relative abundance of *Fibrobacter* genera is present in horses with undifferentiated diarrhea [20]. The Fibrobacteraceae family is essential for the degradation of plant cell wall (PCW) polysaccharides into short-chain fatty acids in the horse’s hind gut. Short-chain fatty acids (e.g., acetate, propionate and butyrate) are a major source of energy for the host, have trophic effects for the colonocytes and play an important role in modulating the immune response of the gastrointestinal tract [30,31]. Furthermore, loss of taxa able to degrade PCW polysaccharides (e.g., Fibrobacteraceae family) can facilitate expansion of bacteria that metabolize starch (e.g., *Lactobacillus*, *Streptococcus* and Lachnospiraceae) [20,32] leading to a greater production of lactate and therefore decrease in luminal pH. Reduction in colonic pH is associated with damage to the intestinal mucosa and a decline in commensal bacteria (e.g., *Escherichia* spp.) eliciting an inflammatory response [33,34,35]. Together, these findings highlight the detrimental effect that antimicrobial drugs have on bacterial communities associated with gut health and indicate that changes in specific taxa could predispose horses to gastrointestinal inflammation and the development of diarrhea.

The increase in Clostridia populations and the Lachnospiraceae family from day 0 to days 3 and 5 in ceftiofur and enrofloxacin treated horses is of interest because genera belonging to these taxa (e.g., *Lachnospira, Roseburia, Butyrivibrio, Eubacterium, Ruminococcus* and *Blautia* genera) play a crucial role in maintaining homeostasis of the gut [21,36,37,38,39]. In our study, commensal Clostridia could have proliferated, but since absolute counting was not performed in the present study, it is possible that the increase was caused because of the reduction in other taxa (i.e., Fibrobacteres and Bacteroidetes). The vast majority of Clostridia present in the equine gut are commensals; however, this genus also includes several species associated with gastrointestinal diseases in horses such as *Clostridium perfringens* and *Clostridiodes difficile* [40]. The experimental design (i.e., lack of testing for enteropathogens) and the low sequence depth prevented us from determining whether a proliferation of pathogenic Clostridia occurred. Nonetheless, this is an important observation considering that many clinicians initiate antimicrobial therapy when treating diarrheic horses and this practice can decrease the relative abundance of important commensals in the GI tract.

The relative abundance of the *Fibrobacter* genus and Clostridia class remained similar in horses treated with oxytetracycline or saline during the study period. This finding was unexpected because reports published during the 1970s and 1980s, but not recently, suggest that administration of intravenous or oral oxytetracycline to horses carry a higher risk for development of AAD than other antimicrobial drugs [41,42,43]. Oral administration of oxytetracycline (10 mg/kg q24h for 5 days or 40 mg/kg q24 for 2 days) to horses caused mild diarrhea associated with an expansion of Enterobacteriaceae, *C. perfringens* type A, *Bacteroides* and *Streptococcus* and loss of *Veillonella* genera [44]. Dissimilarities between studies can be explained by the administered dose of antimicrobials, different routes of administration or differences in methodologies used to investigate the fecal microbiota (i.e., culture vs. DNA sequencing methods).

Alpha diversity richness analysis suggested that numerous low abundance genera, detectable in saline samples, were undetectable on day 3 after antimicrobial treatment. Similarly, administration of trimethoprim and sulfadiazine (TMS, 30 mg/kg) for 5 days orally q12h significantly reduces richness but not diversity, suggesting that antimicrobial drugs affect mainly the low abundance taxa [13]. In our study, microbiota analysis did not include low abundance bacteria <0.05% because of the small sample size, low sequencing depth and the platform-dependent sequencing errors that could result in misclassification of reads (e.g., spurious OTUs that inflated measurements of diversity) [45]. Therefore, studies surveying a larger number of horses per group and a greater sequencing depth are necessary to determine the importance of these low abundance taxa in horses treated with antimicrobial drugs. Conversely, the technology used for microbiota analysis (pyrosequencing) allows sequencing of larger DNA fragments than most of the current studies using Illumina sequencing, increasing confidence of taxonomic classification.

The small number of horses and inclusion of several treatment groups with known interindividual variability of gut microbiota could have increased type II error and decreased statistical power preventing us from identifying further differences between treatment groups [46,47]. Nonetheless, our study offered further evidence that antimicrobial therapy can negatively impact the gastrointestinal microbiota of horses, reduce abundance of GI commensal organisms, and these changes could predispose them to clinically relevant dysbiosis, development of intestinal inflammation and therefore diarrhea or other, subtler subclinical syndromes.

Despite its limitations, studies like this represent an advancement to better understand the effects of exogenous compounds, disease, nutrition, drugs, supplements, prebiotics and probiotics on the equine gastrointestinal microbiome. Information regarding the effects of drugs commonly used in equine practice on the mucosal microbiota is scarce. Implementation of therapies that include the use of nutritional supplements, prebiotics, probiotics and fecal microbial transplantation are controversial because of the paucity of evidence of their benefits. While anecdotal reports and some studies have claimed clinical improvement of various equine gastrointestinal disorders in response to these interventions, the majority have failed to critically demonstrate benefits. These inconsistencies have been the result of inappropriate experimental design, lack of proper controls and inadequate quality and quantity of prebiotics, probiotics and volume of fecal microbial transplants. Additionally, we have learned from numerous microbiome studies in humans that dysbiosis secondary to antibiotic therapy can have long-term detrimental effects. These factors should be considered in the design of future studies as manipulation of the microbiota could potentially improve systemic health and reduce the use of drugs that could be damaging to other body systems.

## 6. Conclusions

This study showed that antimicrobials commonly used in equine practice (ceftiofur, enrofloxacin) can alter equine gastrointestinal communities, which could lead to gastrointestinal disturbances, promote inflammation and development of diarrhea, further antimicrobial resistance and can influence systemic health. By enhancing our understanding of the effect of antimicrobial drugs on equine gut health, it is a major goal to promote the rational use of these and other drugs, but also implement better antimicrobial stewardship practices.

## Figures and Tables

**Figure 1 animals-12-01013-f001:**
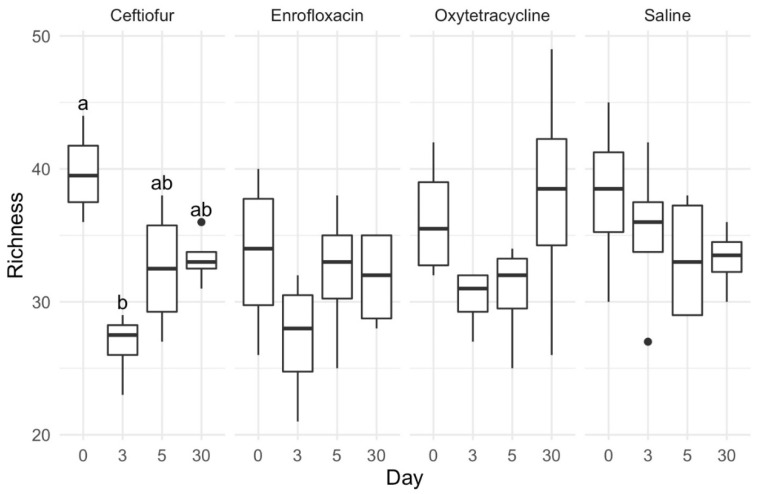
Boxplots of operational taxonomic unit richness throughout treatment duration. Richness was calculated as the total number of genera present in each sample. Data presented as median and interquartile range (IQR). Whiskers are 1.5 × IQR and data outside the whiskers are shown as outliers (black circles). Different letters within a treatment panel indicate significant differences (*p* < 0.05).

**Figure 2 animals-12-01013-f002:**
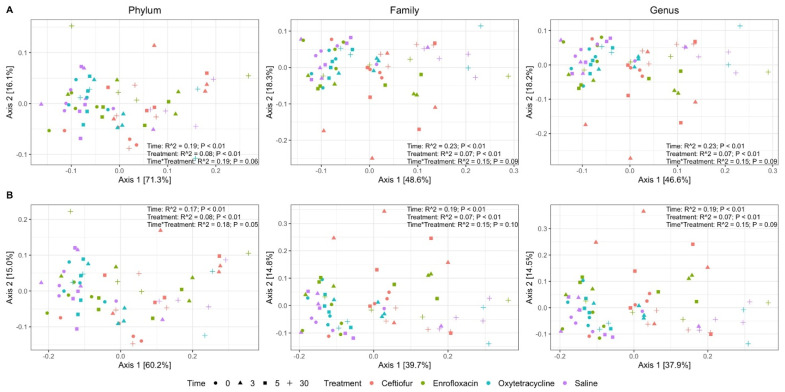
Principal coordinate analysis on Bray-Curtis and Jaccard distances. Distance metrics by panel row are (**A**) Bray-Curtis and (**B**) Jaccard. Samples are colored by treatment and shaped by day. Figure panels in each column are grouped to the displayed taxonomic level. Text within each panel represents the results of PERMANOVA testing against time, treatment and their interactions.

**Figure 3 animals-12-01013-f003:**
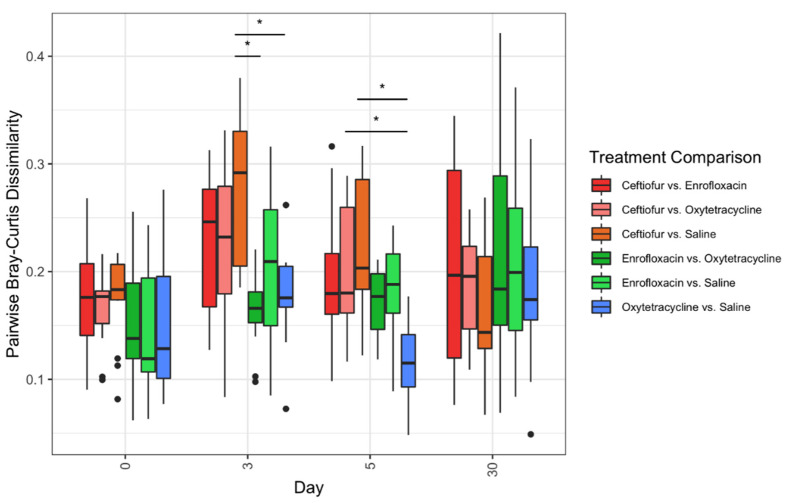
Pairwise Bray-Curtis dissimilarities between treatments by day. Bray-Curtis dissimilarities were measured at genus level taxonomic grouping and stratified by day. Data presented as median and interquartile range (IQR). Whiskers are 1.5 × IQR and data outside the whiskers are shown as outliers (black circles). Asterisks indicate statistically significant differences (*p* < 0.05) within the same day.

**Figure 4 animals-12-01013-f004:**
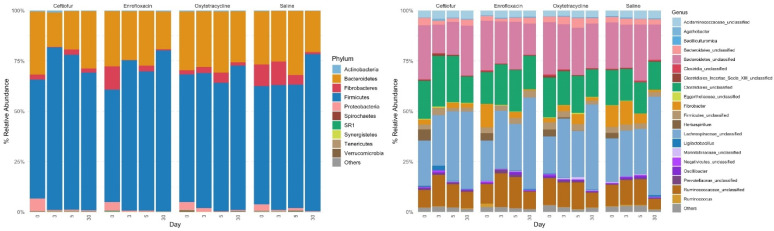
Microbial relative abundances for each treatment group by day. Bacteria were either grouped at phylum or genus level taxonomy. Only top 10 relatively abundant phyla and top 20 relatively abundant genera are shown.

**Figure 5 animals-12-01013-f005:**
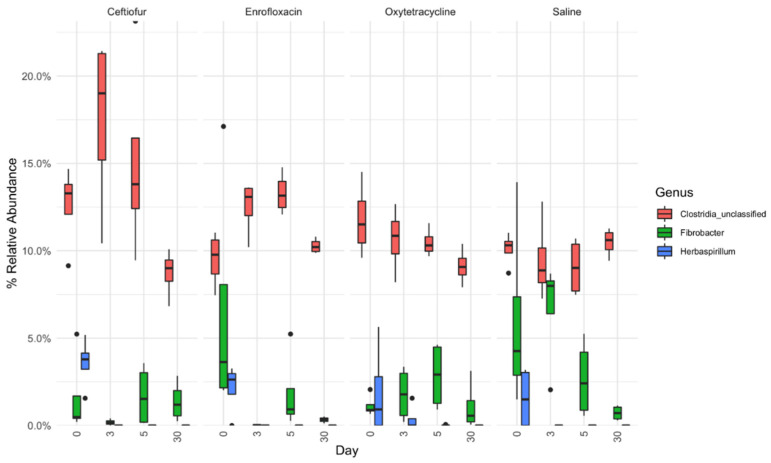
Boxplot relative abundances of three genera throughout treatment duration. Data presented as median and interquartile range (IQR). Whiskers are 1.5 × IQR and data outside the whiskers are shown as outliers (black circles). Relative abundances are shown for *Fibrobacter*, *Herbaspirillum* and unclassified Clostridia. Clostridia_unclassified represents cumulative relative abundance of Clostridia_unclassified, Clostridiales_Incertae_Sedis_XIII_unclassified and Clostridiales_unclassfied as displayed in Figure 4.

## Data Availability

Data supporting the results have been deposited in the Open Science Framework: https://doi.org/10.17605/osf.io/r2dcw.

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
