# Peer review of "Effects of Intravenous Antimicrobial Drugs on the Equine Fecal Microbiome"

_animals, 2022, doi:10.3390/ani12081013_

Round 1
Reviewer 1 Report
This is a well written study about the effects of selected intravenously administered antimicrobials on the microbiota of healthy horses. It is a well written study that adds important information to the existing literature. I only have minor comments
Material and Methods
Do the horses belong to a university research/teaching herd or were they privately owned?
Please more specific about 'no history about antimicrobial administration' - how long have you known the medical record of these horses? Maybe it would be better to state the amount of time that the horses definitely did not receive any antimicrobials
How were the horses managed and fed before the 3 weeks that you describe as standardized fed/management? Were the always together as this group (research/teaching herd) or bought for this study from completely different sources?
You state that all horses were treated during the same time of the year - were they all treated during the same days or one group after another? How much time between groups?
Results
Figure 1, 3, 5 - please indicate what is shown with Boxplots and whiskers (median, range, IQR etc)
Figure 2 and 4 have bad quality
Figure 3 please modify the shades of red as they all appear to be the same
Author Response
This is a well written study about the effects of selected intravenously administered antimicrobials on the microbiota of healthy horses. It is a well written study that adds important information to the existing literature. I only have minor comments
RESPONSE: Thank you for your positive comments
Material and Methods
Do the horses belong to a university research/teaching herd or were they privately owned?
RESPONSE: Horses were university-owned. This was clarified in the manuscript.
Please more specific about 'no history about antimicrobial administration' - how long have you known the medical record of these horses? Maybe it would be better to state the amount of time that the horses definitely did not receive any antimicrobials
RESPONSE: The comment is valid. We know the complete medical history of the horses since they are owned by the university and we have complete medical records. A sentence was added that the horses had no history of antimicrobial administration for at least 6 months prior to the study.
How were the horses managed and fed before the 3 weeks that you describe as standardized fed/management? Were the always together as this group (research/teaching herd) or bought for this study from completely different sources?
RESPONSE: Horses were together at this location, belonged to The Ohio State University and were not purchased for the purpose of this study. All horses in the herd, including the ones in the study, had free access to a standard diet of grass day
You state that all horses were treated during the same time of the year - were they all treated during the same days or one group after another? How much time between groups?
RESPONSE: Horses were treated the same time of the year (same days). This was further clarified in this section of the manuscript.
Results
Figure 1, 3, 5 - please indicate what is shown with Boxplots and whiskers (median, range, IQR etc.)
RESPONSE: Box plots show median, IQR, whiskers are 1.5*IQR, data outside the whiskers are shown as outlier points. This was added to the manuscript.
Figure 2 and 4 have bad quality
RESPONSE: This is due to Word's tendency to compress large image files. Figures within the document were replace with better ones. High resolution figure files can be found in the following link:
https://drive.google.com/drive/folders/1C_gqDd6hER8tcLln3iRlwBKogVbyDaFE?usp=sharing
Figure 3 please modify the shades of red as they all appear to be the same
RESPONSE: As above. Changed left boxplot to dark red, middle box to light red/pink, and right box to orange.
Reviewer 2 Report
The paper Liepman et al. is really interesting, well-designed, suitably presented and discussed. I suggest only some minor changes:
Lines 64-72: please, provide ad hoc relevant references.
Lines 76-79: I suggest deleting the “hypothesized” scenario.
Lines 86-91: please, add more detail about these points.
Line 107: please, add more detail about these points.
Lines 167-171: please, add more detail about these points.
Author Response
The paper Liepman et al. is really interesting, well-designed, suitably presented and discussed. I suggest only some minor changes:
RESPONSE: Thank you for your comments/suggestions.
Lines 64-72: please, provide ad hoc relevant references.
RESPONSE: References added – review article by Barr et al. #4; Costa et al. #13
Lines 76-79: I suggest deleting the “hypothesized” scenario.
RESPONSE: We would prefer to leave the “hypothesis” since that’s what we were anticipating.
Lines 86-91: please, add more detail about these points.
RESPONSE: Valid suggestion and additional details about the horses and their management have been included. As noted for reviewer 1, horses were university-owned, were housed together, under a regular deworming and vaccination program, receiving the same diet, and had not history of antimicrobial administration for at least 6 months prior to the study. Additional details were included in this section (red text).
Line 107: please, add more detail about these points.
RESPONSE: Information in this section is considered standard for equine clinical studies and rarely included. However, we added additional details.
Lines 167-171: please, add more detail about these points.
RESPONSE: Information in this section is considered standard for equine clinical studies and rarely included. Since there were no changes in fecal consistency we did not think it was necessary to add more details in this section.
Reviewer 3 Report
In this paper, Liepman and colleagues address the issue of side effects resulting from the administration of antimicrobial drugs in horses. With the help of 16S rRNA sequencing the study rteveales fluctuations particularly in the beta diversity and bacterial abundance of the gut microflora.
While several reports have indicated changes in the fecal microbiome following antibiotic trreatment, this study takes advantage of modern molecular techniques and strives to add physical examination and blood analysis to provide a more complex view on the topic. Although according to the authors, no changes in physical examination parameters, hematology or biochemical indices, these data are not shown in the manuscript (perhaps these could be published as supplementary materials).
I believe the study is well designed and executed, the discussion could be amplified by taking into consideration nutritional supplements, prebiotics or probiotics that could stabilize the GIT and/or fecal microbiome.
On a minor note, the references in the main body should be written in brackets rather than superscripts.
Author Response
While several reports have indicated changes in the fecal microbiome following antibiotic treatment, this study takes advantage of modern molecular techniques and strives to add physical examination and blood analysis to provide a more complex view on the topic. Although according to the authors, no changes in physical examination parameters, hematology or biochemical indices, these data are not shown in the manuscript (perhaps these could be published as supplementary materials).
RESPONSE: While the point about including physical examination data is valid, this is not customary for this type of clinical study, when the focus of the study was not on clinical variables which did not change during the study period.
I believe the study is well designed and executed, the discussion could be amplified by taking into consideration nutritional supplements, prebiotics or probiotics that could stabilize the GIT and/or fecal microbiome.
RESPONSE: We added a section at the end of the discussion on nutritional supplements, prebiotics, probiotics as well as the potential clinical value of fecal microbial transplantation (in red).
On a minor note, the references in the main body should be written in brackets rather than superscripts.
RESPONSE: Valid suggestion – references have been changed to Animals format in brackets.